

# Estimates of tropical cyclone geometry parameters based on best track data

Kees Nederhoff[1], Alessio Giardino[1], Maarten van Ormondt[1], Deepak Vatvani[1]

[1]Deltares, Marine and Coastal Systems, Boussinesqweg 1, 2629 HV Delft, The Netherlands

*Correspondence to*: Kees Nederhoff (kees.nederhoff@deltares.nl)

**Abstract.** Parametric wind profiles are commonly applied in a number of engineering applications for the generation of tropical cyclone (TC) wind and pressure fields. Nevertheless, existing formulations for computing wind fields often lack the required accuracy when the TC geometry is not known. This may affect the accuracy of the computed impacts generated by these winds. In this paper, empirical stochastic relationships are derived to describe two important parameters affecting the
TC geometry: radius of maximum winds (RMW) and the radius of gale force winds (ΔAR35). These relationships are formulated using best track data (BTD) for all seven ocean basins (Atlantic, S/NW/NE Pacific, N/SW/SE Indian Oceans). This makes it possible to a) estimate RMW and ΔAR35 when these properties are not known and b) generate improved parametric wind fields for all oceanic basins. Validation results show how the proposed relationships allow the TC geometry to be represented with higher accuracy than when using relationships available from literature. Outer wind speeds can be
well reproduced by the commonly used Holland wind profile when calibrated using information either from best-track-data or from the proposed relationships. The scripts to compute the TC geometry and the outer wind speed are freely available via the following URL.

## 1 Introduction

Tropical cyclones (TCs) are among the most destructive natural hazards worldwide. TCs can cause hazardous
weather conditions including extreme rainfall and wind speeds, leading to coastal hazards, such as extreme storm surge levels and wave conditions. The impact of TCs are different in developed and developing countries. Generally, the worst effects in the developed world are direct economic losses. In the United States (U.S.) alone, the mean annual damage due to TCs was estimated by Willoughby (2012) as 11.0 billion USD (year 2015). In the developing world, TCs conflict in immense social costs in terms of destruction and mortality. For example, between 1960-2004 more than half a million
inhabitants of Bangladesh died as a consequence of TCs, primarily due to storm surge (Schultz et al., 2005). Additionally, TCs can also have devastating effects on nature, geomorphology, agriculture and fresh water supply. Thus, due to the extensive costs in lives, property and other damages, the ability to effectively model these storms is essential.

Numerical models can be applied to quantify the effects of TCs (Giardino et al., 2018). In hindcasting studies, this is generally done by using surface winds derived by data assimilation techniques (e.g. HRD Real-time Hurricane Wind



Analysis System or H*WIND; e.g. Powell et al. 1998,).  However, in multi-hazard risk assessments, the spatial distribution of surface winds is generally not known. Therefore, wind fields based on best track data (BTD) or synthetic tracks, are generated using parametric wind profiles. Several (horizontal) parametric wind profiles (e.g. Fujita, 1952; Willoughby et al., 2006, Chavas et al., 2015) exist in literature, with the original Holland wind profile (Holland, 1980; hereafter H80) being the

most widely used due to its relative simplicity. However, without calibration, parametric wind profiles are often unable to accurately reproduce the spatial distribution of winds in TCs (e.g. Willoughby and Rahn, 2004). This potentially leads to an under- or overestimation of wind speeds and associated coastal hazards. Calibration of TC formulations is possible by applying additional relationships, supported by the use of suitable data. In particular, information on the wind radii of cyclones can constrain the decay of wind speeds away from the eye wall and can be included in the most recent version of

the Holland wind profile formulation (Holland et al., 2010; hereafter H10).

The radius of maximum winds (RMW), which describes the distance from the center to the strongest axially symmetric wind in the core of the cyclone, is one of the most important parameters to define a parametric wind profile. Moreover, the RMW plays an important role in the assessment of hazards induced by TCs since the storm surge level increases as a function of the RMW (Loder et al., 2009). Several relationships exist in literature to estimate the RMW (e.g.

Willoughby et al., 2006; Vickery & Wadhera, 2008; Knaff et al., 2015). However, these relationships are derived either for the Atlantic and/or Eastern Pacific Ocean (i.e. U.S. coast) and are therefore not necessarily valid for other ocean basins. Each ocean basin has its own climatological properties and, for example, there seems to be a observational relationship between (mean) storm size, in terms of precipitation area (Lin et al., 2015) or wind speeds (Chavas et al., 2016), and the relative sea surface temperature (SST). The reason that most relationships are derived for the U.S. coast is because of the high-quality

data availability (i.e. aircraft reconnaissance data). Relationships that estimate wind radii at different wind speeds are scarcer. Knaff et al. (2007) describe explicitly the TC surface winds using a modified Rankine vortex, which makes it also possible to compute different wind radii corresponding to different wind speeds (i.e. 34, 50, 64, 100 kts). However, these results are derived from BTD of the Atlantic, Northeast Pacific and Northwest Pacific Oceans.

In the last decades, a large amount of higher quality data has become available which can be used to improve and

validate the relationships and parametric wind profiles found in literature. In addition to the RMW, the wind radii of 35 (or 34), 50, 65 (or 64) and 100 knots (hereafter referred to as R35, R50, R65, R100) for the four geographical quadrants around the cyclone are currently recorded (see also left panel of Figure 1). There are numerous sources that can provide information on the spatial distribution of surface winds ranging from in-situ observations (e.g. surface reports and buoy observations) to scatterometry (e.g. QuikSCAT, see Chavas & Vigh, 2014). Some methods are more reliable than others, but a posteriori it is

not clear which sources were used for individual wind radii estimates in the best-track data (BTD). However, the currently operationally available satellite-based wind radii estimates are characterized by higher accuracy than in the past (Sampson et al., 2017).

In this paper, new relationships are proposed to estimate the median RMW and radius of gale force winds (ΔAR35) for each ocean basin. In addition, the standard deviation of the TC geometry is described explicitly, making it possible to





treat the TC geometry stochastically with a certain probability distribution. This means that TC geometry is a random variable whose possible values are an ensemble of different outcomes. This is useful when TC size is not known and the probability of a relatively large and/or small TC and consequent risks needs to be assessed (e.g. in a Monte Carlo analysis with synthetic tracks). Moreover, the paper demonstrates how the proposed relationships lead to improved error statistics

compared to those found in literature. On top of that, validation with QSCAT-R shows that outer wind speeds can be well reproduced by a parametric wind profile while using the newly developed relationships or observed values for RMW and wind radii.

This paper is outlined as follows: Section 2 describes the data used for the study. The new relationships describing the radius of maximum winds and radius of gale force winds are derived in Section 3 and then validated in Section 4.

Finally, Section 5 and 6 discuss and summarize the main conclusions of the study.

## 2. Data

### 2.1 Best track data (BTD)

Two data sources were used to describe the RMW and R35: data from the North Atlantic and Northeast and North-Central Pacific dataset from the National Hurricane Center (NHC) and the dataset from the Joint Typhoon Warning Center

(JTWC). The second dataset includes data from different ocean basins (Northwest Pacific Ocean, the South Pacific Ocean and Indian Ocean). Note that the estimation of wind radii is rather subjective and strongly dependent on data availability as well as different climatology and analysis methods (e.g. aircraft observations versus the Dvorak method). In this paper, all the available data were used and potential shortcomings in the data are disregarded in order to fit new empirical stochastic relationships with the largest possible dataset and for every ocean basin separately. This approach, with its advantages and

disadvantages, is discussed in Section 5.1. Some of the historical records do not contain values for either the RMW or R35 and therefore these records are discarded. Although these BTD are used as ground truth, the errors in the best-track wind radii are estimated to be as high as 10%–40% (e.g. Knaff and Sampson 2015). The accuracy of a single record depends on the quality and quantity of the available observational data. For example, if in situ observations were available in proximity to the TC or if a complete scatterometer passed over the TC, the accuracy may increase. However, information on the

accuracy is not available per single data entry.

The archives from the NHC and JTWC contain six-hourly storm positions and maximum intensity estimates of tropical and subtropical systems. For this analysis, all data points with wind speed of 20 m/s or higher were included in the study, since the focus is on tropical storms. Moreover, it is expected that parametric wind profiles cannot capture subtropical systems. Also, data points with an RMW larger than 100 kilometers (km) were excluded from the analysis because,

generally, those points refer to tropical depressions, with large spatial coverage, which are outside the scope of this study. Moreover, the averaged value of R35 ($\overline{R35}$) over the four quadrants, similarly to Carrasco et al. (2014), was used. Only data entries with an estimate of R35 for all four quadrants were used. Therefore, all data entries have both an estimate for RMW




and $\overline{R35}$. On top of that, using all the six-hourly storm positions and maximum intensity estimates in the calibration and validation assumes statistical independence.

In this paper, TC geometry variables RMW and R35 were treated as stochastic variables. This means that, although physically not realistic, RMW could assume larger values than R35. In order to overcome this, a new variable was defined:

5    the average difference in radius of 35 knots (ΔAR35; similar to Xu & Wang, 2015), or radius of gale force winds, describing the difference between the RMW and the average radius of 35 knots (AR35), see Equation 1. In practical applications, one would first retrieve the RMW based on data or estimate the RMW based on an empirical relationship. Secondly, the R35 would be calculated by adding up the RMW with the ΔAR35 (see also right panel of Figure 1). An additional advantage of introducing this new variable is that ΔAR35 contains considerable less scatter.

$$\Delta AR35 = AR35 - RMW \quad (1)$$

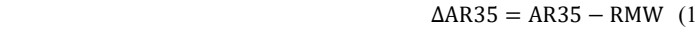

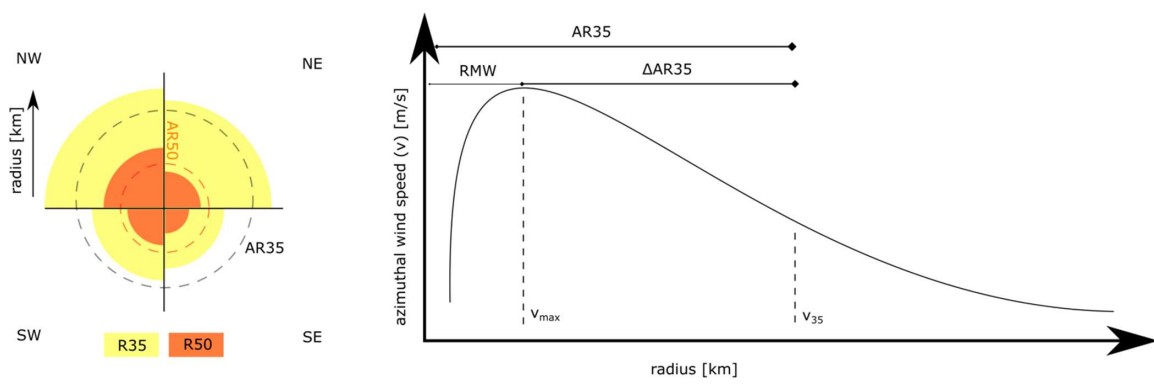

**Figure 1**      **Sketch of the terminal used in this paper. Left panel difference in wind radii from different quadrants (NW, NE, SE, SW). Right panel difference in RMW, AR35 and ΔAR35.**

15    The BTD is divided into a calibration period (2000-2014) and a validation period (2015-2017). The combined BTD from the NHC and JTWC contains a total of 18,903 unique historical TC data entries, of which 14,800 were used for the calibration of the new empirical (stochastic) relationships, and 4,103 for the validation of the estimated wind radii.

**2.2 QSCAT-R**

The QuikSCAT-based QSCAT-R database (Chavas & Vigh, 2014), with data for the period 1999–2008, was used

20    to validate the computed outer (azimuthal) winds using H10 wind profile and the new proposed empirical relationship. The dataset, developed by researchers at the NASA Jet Propulsion Laboratory (JPL), is derived from the latest version of the QuikSCAT near-surface ocean wind vector database. It includes 690 unique TC profiles and it is optimized specifically for


tropical cyclones with higher wind speeds. QuikSCAT measurements are accurate in all weather conditions for winds up to 40 m/s (Stiles et al., 2013), while their precision decreases for the inner wind speeds in the TC core (Hoffman & Leidner, 2005). Therefore, QSCAT-R data were only used to validate the outer wind speeds, and not the inner wind speeds or TC core. The tropical cyclone dataset carries a 1-2 m/s positive bias and a 3 m/s mean absolute error, which are not further

discussed or taken into account in the analysis.

### 2.3 Ocean basins

According to the WMO (World Meteorological Organization), areas of TC formation were divided into seven basins (Figure 2A). These include the North Atlantic Ocean (NAO), the Northwest Pacific Ocean (NWPO), the Northeast Pacific Ocean (NEPO), the South Pacific Ocean (SPO), the Southwest Indian Ocean (SWIO), the Southeast Indian Ocean

(SWEI) and the North Indian Ocean (NIO). Other ocean basins (e.g. the South Atlantic Ocean) were not included in this study since weather systems in these areas rarely form a TC.

### 2.4 Data conversion

Data were converted to International System of Units (SI) units (wind speeds in m/s from knots with a conversion of 1 kt = 0.514 m/s and wind radii in kilometers (km) from nautical mile with a conversion of 1 nm = 1.852 km). Throughout

this study, a maximum cyclone sustained wind $v_{max}$ has been determined at a 10-m elevation over open sea and 1-minute averaged. The reason for this averaging is to be consistent with the JTWC and NHC which also reports the maximum sustained surface winds in terms of 1-minute mean wind speed. Other nations, however, report maximum sustained surface winds averaged over different time intervals, which in some cases is 10 minutes. Also, numerical models often require 10-minute averaged winds. For the conversion of 1-minute to 10-minute averaged wind speed, a conversion factor equal to 0.93

can be used, based on WMO guidelines (Harper et al., 2010). However, in this study, conversions between 1-minute and 10-minute wind speeds were not needed.

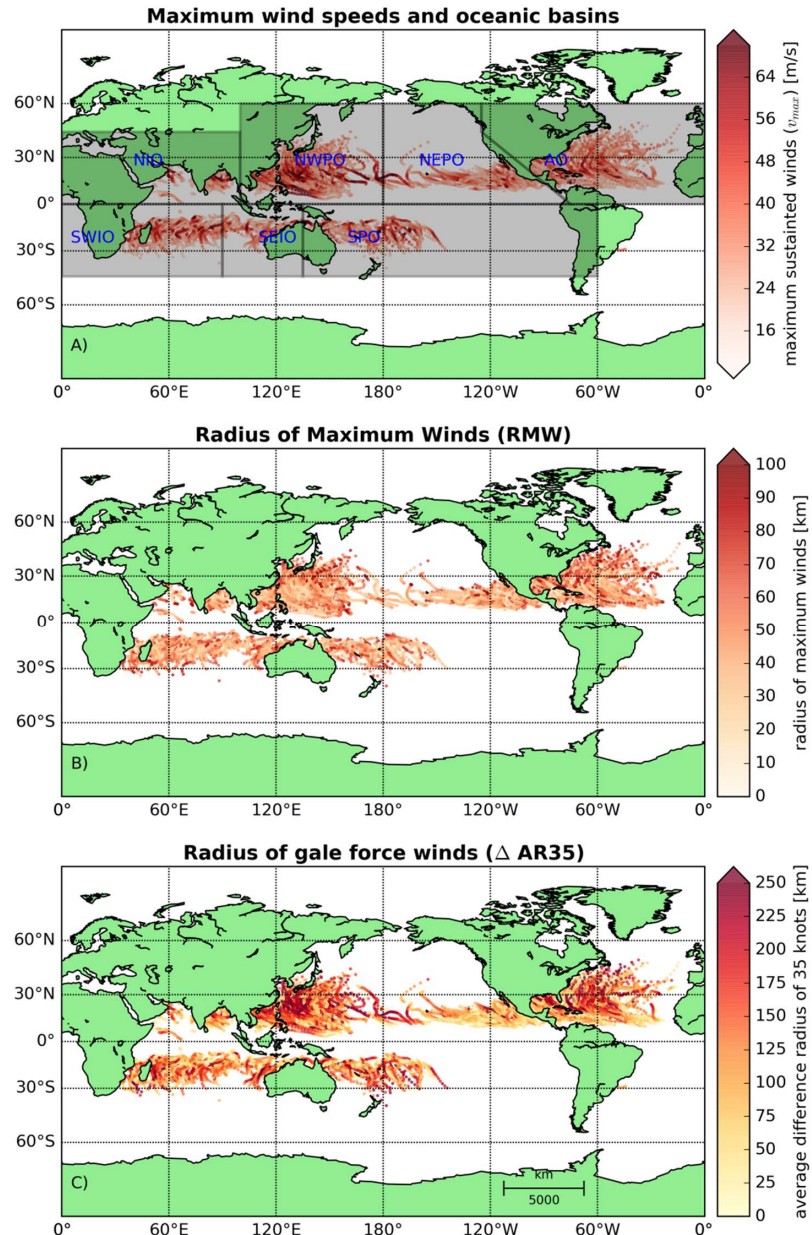

**Figure 2** Maximum sustained wind speeds and definition of the different ocean basins (panel A). Radius of maximum winds (panel B) and radius of gale force winds (panel C) for all the BTD.





### 3. New empirical relationships

In this section, empirical relationships to estimate the radius of maximum winds (RMW; Section 3.1, see Figure 2B) and the radius of gale force winds (ΔAR35; Section 3.2, see Figure 2C) were derived based on BTD from the calibration period (2000-2014).

### 3.1 Radius of maximum winds (RMW)

The Vickery and Wadhera (2008) relationship, derived for all major hurricanes ($\Delta p_c$ > 30 hPa or $v_{max}$ >35 m/s) in the Gulf of Mexico and Atlantic Ocean (hereafter VW08), is one of the several relationships in literature providing an estimate of the RMW. VW08, derived based on H*WIND data, relates RMW to pressure drop in the eye and latitude. While

we acknowledge the existence of several other relationships to estimate the RMW, VW08 was used due to its relative simplicity. Figure 3 compares RMW data from the BTD during the calibration period with results from VW08 in the form of a scatter plot with the maximum sustained wind speed ($v_{max}$) indicated by color intensity. The data shows a large amount of scatter, both for lower and higher RMW values. However, there is a clear pattern visible that larger maximum sustained wind speeds result in a smaller RMW. This is in line with other observations (e.g. Willoughby et al., 2016) or based upon

idealized Sawyer Eliassen models (e.g. Schubert and Hack, 1982; Willoughby et al., 1982) that TC eyewalls generally contract during intensification. There is also a tendency in the dataset for TCs at higher latitudes to have larger eye diameters (e.g. Knaff et al., 2015; not shown here). The large negative bias of 17 km, computed as a difference between observed and computed RMW is noteworthy, indicating that VW08 often underestimates the RMW, especially for lower maximum sustained wind speeds.

Given the large spread in the data, as also shown in Figure 3, it was decided to treat RMW as stochastic variable. Instead of directly deriving an empirical equation which relates RMW to $v_{max}$ using a least-square fitting procedure as typically done in similar studies, the following approach was used. At first, parameters of a probability density function (PDF) that fits the variation of RMW for a range of $v_{max}$ and latitude values were fitted. Then empirical equations were derived that relate these parameters to $v_{max}$ and latitude. The benefit of this approach is that it can produce an estimate of the

most probable value for RMW (i.e. mode) or median/mean as well as its variance (e.g. 90% prediction interval, PI).

First, the RMW for each TC category were fitted to various parent distributions. In particular, the following fitting parent distributions were tested by visual comparison and by applying the Kolmogorov-Smirnov test: normal, lognormal, Gumbel, Rayleigh and gamma. The lognormal distribution was found to provide the best fit with the measured data, and therefore further used to describe the distribution of RMW. This is also consistent with the distribution used for describing

ΔAR35 and findings in literature (e.g. Dean et al., 2009). Secondly, the chosen parent distribution was used to fit the BTD in order to derive shape ($\sigma$) and location parameter ($\mu$) of the lognormal distribution, dependent on latitude and wind speed. In particular, the BTD from the calibration period were divided based on a moving window with a bin of 10 m/s in wind speed




and of 10 degrees in latitude (0-10, 1-11, 2-12, etc.). A shape parameter was used with an exponential decay function, and fitting coefficients constant per each ocean basin. This resulted in Equation 2 for the $\mu$ parameter which, for a log-normal distribution, corresponds to the median value:

$$\mu_{RMW} = A_2 e^{-\frac{v_{max}}{B_2}}(1 + C_2|\theta|) + D_2 \qquad (2)$$

where $\mu_{RMW}$ represents the location parameter of the lognormal distribution for RMW, $v_{max}$ is the maximum (1-minute averaged) wind speeds, $\theta$ is the latitude in degrees, and $A_2$, $B_2$, $C_2$ and $D_2$ are fitting coefficients.

      As observed in literature (e.g. Knaff et al. 2015), the median RMW ($\mu_{RMW}$) in Equation 2 depends on $v_{max}$ (i.e. higher wind speeds result in lower RMW) and latitude (i.e. higher latitude result in higher RMW). The addition of storm

duration or the use of the axisymmetric component of the wind speed only as input parameters resulted in very limited skill improvement in the estimation of RMW; therefore these variables were discarded. This procedure was applied to the combined JTWC and NHC BTD from the calibration period at all basins, and then for each individual ocean basin. Table 1 contains the shape and location values for the fitting parameters to be used in Equation 2.

      A scatter plot describing the RMW derived from BTD as a function of the maximum wind speed and for (an

arbitrarily chosen) latitude of 10 degrees and computed as according to Equation 2, is shown in Figure 4. The green line shows the median RMW based on the BTD, whereas the solid blue line represents the mean RMW obtained from Equation 2. The black lines indicate the 5 and 95 percent exceedance values computed based on BTD. Finally, the 90% prediction interval, is shown using the filled red color. The figure shows how the variance in RMW decreases (both in the data as in the empirical relationship) as a function of $v_{max}$, indicating that faster-rotating cyclones are characterized by less noise. The new

empirical equation for RMW is evaluated in Section 4.

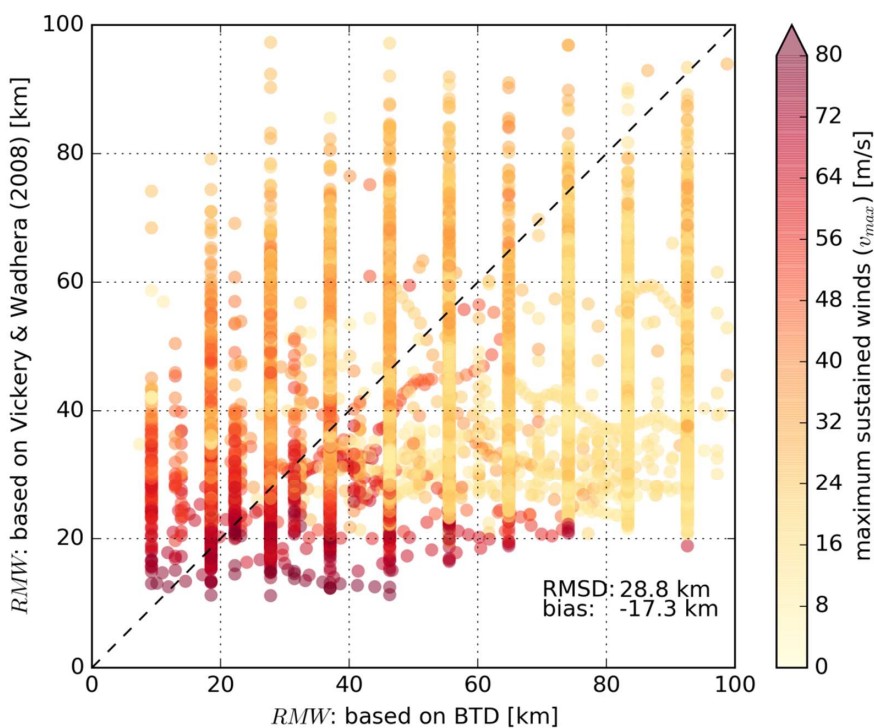

**Figure 3**          **Scatter plot describing BTD RMW versus computed RMW based on VW08. Data points are colored-coded based on the maximum sustained wind speeds in the BTD. The dashed line represents a perfect fit between BTD and computed data based on VW08.**

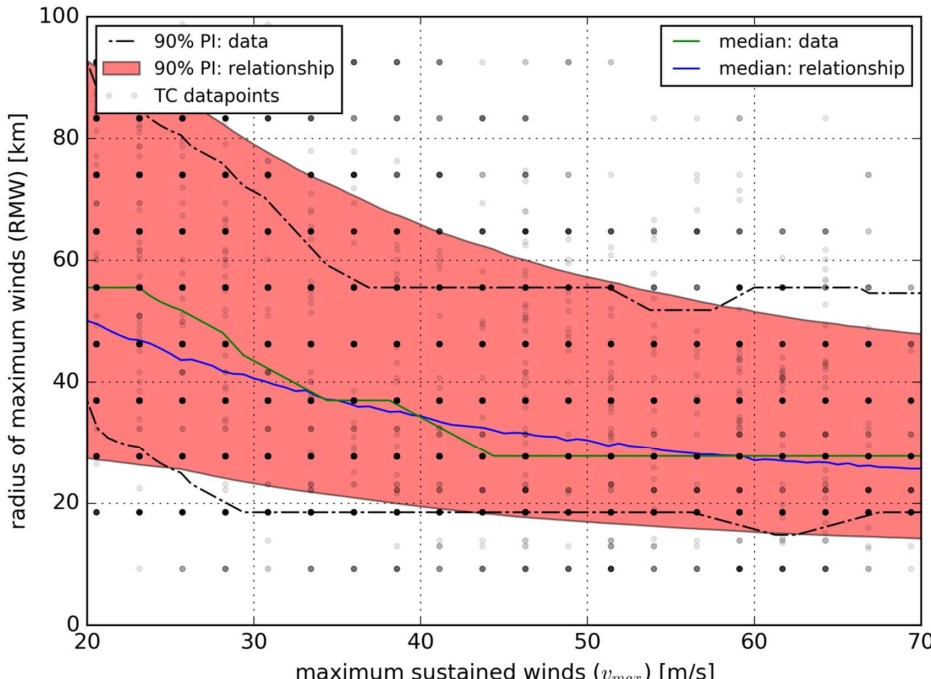

**Figure 4** **Scatter plot describing RMW (BTD and computed) as function of the maximum sustained wind speeds (and the latitude; not shown). The blue line is the median of the proposed relationship derived for all basins at an arbitrarily chosen latitude of 10°. The green line is the median of the BTD. The red area shows the 90 percent prediction interval (PI) based on the proposed relationship for all basins. The 5 and 95 percent exceedance values from the BTD are presented as black dashed lines. Gray dots are observation points. More frequent observations are shown as darker points.**



Table 1        Fitting coefficients for the lognormal RMW as described in Equation 2.

|  | shape | Location |  |  |  |  |
|---|---|---|---|---|---|---|
| Basin | A1 | A2 | B2 | C2 | D2 | Count |
| NIO | 0.307 | 132.4 | 14.6 | -0.003 | 20.4 | 480 |
| SWIO | 0.338 | 229.2 | 9.5 | 0.004 | 28.4 | 1889 |
| SEIO | 0.343 | 85.3 | 30.7 | 0.002 | 5.8 | 832 |
| SPO | 0.364 | 127.8 | 11.8 | 0.016 | 25.5 | 1118 |
| NWPO | 0.359 | 153.7 | 11.5 | 0.007 | 28.9 | 4836 |
| NEPO | 0.311 | 261.5 | 7.0 | 0.026 | 29.2 | 2570 |
| AO | 0.395 | 19.1 | 24.1 | 0.106 | 23.2 | 3075 |
| All | 0.370 | 44.8 | 23.4 | 0.030 | 22.4 | 14800 |


### 3.2 Radius of gale force winds (ΔAR35)

By applying a parametric wind profile, it is possible to derive the ΔAR35. Here, the H10 wind profile was applied, in which the B parameter was computed based on H80 (Equation 3A), and in which information on the wind radii of cyclones was used to constrain the decay of wind speeds away from the eye wall (Equation 3B). When no additional

information on the wind radii is provided, H10 reduces to the original H80 wind profile which is often unable to accurately reproduce the spatial distribution of winds in TCs (e.g. Willoughby and Rahn, 2004).

$$B = \frac{v_{max}^2 \rho_a e}{100(\Delta p_c)} \qquad (3A)$$

$$x = 0.5 + (r - RMW)\frac{x_n - 0.5}{r_n - RMW} \qquad (3B)$$

where B represents the Holland pressure profile parameter, $\rho_a$ is the air density (assumed constant at 1.15 kg/m$^3$), $e$ is the base of natural logarithms, $\Delta p_c$ is the pressure drop in the core of the TC in hPa, $x$ is the exponent used to compute the wind profile in H80/H10 and $x_n$ represents the adjusted exponent to fit the peripheral observations at radius $r_n$.

Knaff et al. (2007) relationships (hereafter CLIPER; CLImatology and PERsistence models), derived for the NAO,

NWPO, and NEPO are among the few in literature providing an estimate of the TC surface winds. Knaff et al. (2007) fitted a modified Rankine vortex on the BTD of NHC and JTWC, which makes it possible also to retrieve the ΔAR35. Figure 5 compares ΔAR35 from the BTD, derived from the calibration period, with results from CLIPER, in which v$_{max}$ is indicated by color intensity in the scatter plot. The data show a large amount of scatter and bias. However, there is a clear pattern showing that larger maximum sustained wind speeds result in a larger ΔAR35. There is also a tendency in the dataset for

TCs at higher latitudes to have a larger ΔAR35 (not shown here).

In order to improve the estimate of the ΔAR35, generic relations were derived as part of this study based on BTD from the calibration period from all ocean basins, as well as data from each individual basin separately. The method followed is similar to the one applied to estimate RMW. First, a representative parent distribution of the data was sought for, secondly, the parameters of the PDF were determined and thirdly, the parameters of the PDF were fitted for a range of v$_{max}$

and latitude values. The same parent distributions were tested and the lognormal distribution was again chosen as most representative, which is in line with Chavas et al. (2016).

Similarly to RMW, the BTD from the calibration period were divided based on a moving window with a bin width of 10 m/s in wind speed (0-10, 1-11, 2-12, etc.) and of 10 degrees in latitude. This led to Equation 4 in which exponential functions, dependent on the wind speed per oceanic basin, were used to describe the location parameter and the shape

parameter. Additionally, the analysis of the data showed that ΔAR35 is dependent on the latitude, with TCs generally increasing in size at higher latitudes. Adding additional parameters (e.g. storm duration or intensity change of the wind speed) resulted in very limited skill improvement for the estimate of ΔAR35. This procedure was applied to both the





combined JTWC and NHC BTD from the calibration period of all basins, and for each individual ocean basin. Table 2 contains the values for the fitting parameters for the ΔAR35 of Equation 4.

$$\mu_{AR35} = A_3 + e^{v_{max} \cdot B_3} \cdot (1 + C_3 |\theta|) \qquad (4)$$

$$\sigma_{AR35} = A_4 + (v_{max} - 18)^{B_4} \cdot (1 + C_4 |\theta|)$$

where $\mu_{AR35}$ and $\sigma_{AR35}$ represent, respectively, the location and shape parameter of the lognormal distribution for AR35 and A3, A4, B3, B4, C3 and C4 are fitting coefficients.

10      A scatter plot describing the ΔAR35 derived from BTD as a function of the $v_{max}$ and latitude and computed as according to Equation 4 is shown in Figure 6. The green line shows the median ΔAR35 based on the BTD, whereas the solid blue line represents the mean ΔAR35 obtained from Equation 4. The black lines indicate the 5 and 95 percent exceedance values computed based on BTD. Finally, the 90% prediction interval is shown using a filled red color. The figure shows how the median ΔAR35 increases as a function of $v_{max}$ while the variance stays fairly constant. The new empirical equation for

15   ΔAR35 is evaluated in the next chapter.

**Table 2**      **Fitting coefficients for the lognormal RMW as described in Equation 4.**

|  | **shape** |  |  | **location** |  |  |  |
|---|---|---|---|---|---|---|---|
| **Basin** | **A3** | **B3** | **C3** | **A4** | **B4** | **C4** | **Count** |
| NIO | 0.1215 | -0.0522 | 0.0329 | 30.93 | 0.531 | -0.012 | 480 |
| SWIO | 0.1312 | -0.0444 | 0.0023 | 30.21 | 0.415 | 0.022 | 1889 |
| SEIO | 0.1223 | -0.0454 | 0.0133 | 26.59 | 0.426 | 0.029 | 832 |
| SPO | 0.1205 | -0.0350 | -0.0052 | 23.88 | 0.431 | 0.038 | 1118 |
| NWPO | 0.1561 | -0.0417 | 0.0050 | 33.27 | 0.429 | 0.017 | 4836 |
| NEPO | -0.2513 | -0.0091 | -0.0051 | 18.11 | 0.486 | 0.030 | 2570 |
| AO | 0.1319 | -0.0421 | 0.0124 | 17.00 | 0.454 | 0.055 | 3075 |
| All | 0.1900 | -0.0446 | 0.0061 | 29.61 | 0.413 | 0.024 | 14800 |


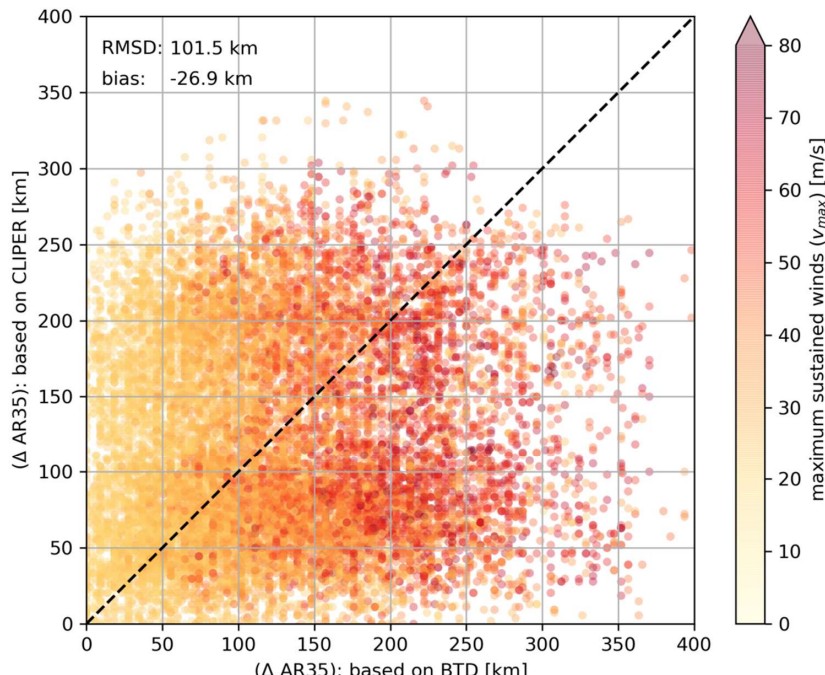

**Figure 5** **Scatter plot describing BTD ΔAR35 versus computed ΔAR35 based on CLIPER. Data points are colored-coded based on the maximum sustained wind speeds in the BTD. The dashed line represents a perfect fit between the BTD and the computed data based on CLIPER.**

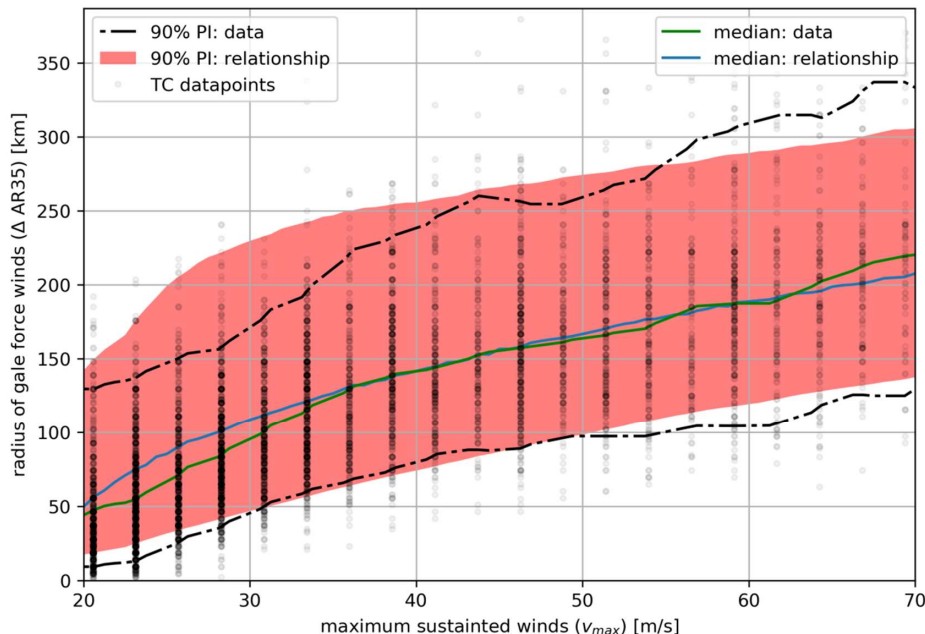

**Figure 6**           **Scatter plot describing ΔAR35 (BTD and predicted) as function of the maximum sustained wind speeds (and the latitude; not shown). The blue line is the median of the proposed relationship derived for all basins at an arbitrarily chosen latitude of 10°. The green line is the median of the BTD. The red area shows the 90 percent prediction interval based on the proposed relationship for the standard deviation. The 5 and 95 percent exceedance values from the BTD are presented as black solid lines. Gray dots are observation points. More frequent observations are shown as darker points.**


**4. Validation**

In this section, empirical relationships to estimate the RMW and ΔAR35 were validated based on BTD from the validation period (2015-2017) (Section 4.1). Moreover, the outer wind profile based on the Holland wind profile, in combination with observed wind radii, were further validated using the QSCAT-R database (Section 4.2).

**4.1 Wind radii**

A subset of the BTD (from 2015-2017) was used to validate the wind radii. Errors statistics are summarized in Table 3. The values indicate that, for all basins combined, the root-mean-square deviation (RMSD) between the BTD and the proposed relations for the RMW is 17 percent lower than compared to VW08 (RMSD of 18 km compared to 21 km). In the NEPO basin, VW08 performs relatively better than at other basins. When comparing the performance of the proposed

relations and VW08, it is important to note that the relation of VW08 was derived for storms with central pressures lower than 980 hPa, thereby explicitly focusing on the most severe TCs. When the data were filtered to include only data points with a pressure drop ($\Delta p_c$) larger than 30 hPa, the RMSD decreases and differences become much smaller (0-10% decrease in RMSD). Moreover, the bias also decreases.

Table 4 shows the error statistics related to the estimation of ΔAR35. In particular, the RMSD between the proposed

relations and the BTD for all basins combined is 25% percent lower compared to CLIPER (RMSD of 74 km compared to 94 km) and there is a negative bias ranging between 9 and 37 km. Remarkably, the deviations of the ΔAR35 based on BTD in the NIO and SEIO from CLIPER are significantly smaller compared to the differences for the AO for which CLIPER was derived. When the H10 wind profile is applied without additional information to compute the decay of wind speeds away from the eye wall (H80), the ΔAR35 is strongly overestimated (overall bias of 177 km).





**Table 3**   Root-mean-square differences (RMSD; first number) and bias (second number) for RMW in kilometers for the validation period for both the proposed relationships as for VW08. Statistics are presented for all data points, as well for data points with a pressure drop ($\Delta p_c$) larger than 30 hPa.

| Basin | proposed all | VW08 All | proposed, $\Delta p_c > 30$ | VW08, $\Delta p_c > 30$ | Count all | Count $\Delta p_c > 30$ |
|---|---|---|---|---|---|---|
| NIO | 20.9 / -14.2 | 25.3 / -17.7 | 14.0 / -4.5 | 14.1 / -2.5 | 146 | 46 |
| SWIO | 16.8 / -7.0 | 20.0 / -8.6 | 10.4 / -0.2 | 9.8 / 0.3 | 365 | 166 |
| SEIO | 17.9 / -10.7 | 24.0 / -14.0 | 9.6 / -1.6 | 10.9 / 4.9 | 107 | 34 |
| SPO | 18.1 / -9.1 | 22.1 / -10.1 | 12.9 / -3.0 | 12.7 / 1.5 | 424 | 184 |
| NWPO | 17.2 / -6.4 | 22.4 / -5.5 | 12.1 / -0.3 | 14.7 / 4.6 | 1389 | 742 |
| NEPO | 16.9 / -8.4 | 17.5 / -6.7 | 13.1 / -4.8 | 11.6 / -1.2 | 1031 | 311 |
| AO | 21.0 / -8.7 | 21.5 / -0.8 | 17.2 / -4.1 | 18.2 / 8.1 | 641 | 291 |
| All | 18.0 / -7.1 | 21.0 / -6.5 | 13.1 / -1.6 | 14.2 / 3.3 | 4103 | 1774 |

5 **Table 4**   Root-mean-square differences (RMSD; first number) and bias (second number) for $\Delta AR35$ in kilometers for the validation period for the proposed relationships, CLIPER (Knaff et al., 2015) and H80 wind profile.

| Basin | Proposed | CLIPER | H80 | count |
|---|---|---|---|---|
| NIO | 48.0 / -17.5 | 51.0 / 3.3 | 275.2 / 221.9 | 146 |
| SWIO | 68.9 / -31.4 | 123.1 / -95.3 | 248.4 / 190.9 | 365 |
| SEIO | 37.2 / -9.0 | 69.0 / -58.3 | 238.8 / 187.7 | 107 |
| SPO | 59.6 / -16.3 | 104 / -74.7 | 267.2 / 214.1 | 424 |
| NWPO | 83.8 / -37.3 | 95.0 / -25.6 | 294.2 / 198.8 | 1389 |
| NEPO | 47.4 / -10.3 | 86.4 / 68.7 | 125.4 / 59.3 | 1031 |
| AO | 90.0 / -26.0 | 116.8 / 7.1 | 552.7 / 252.4 | 641 |
| All | 74.1 / -23.3 | 94.2 / -13.9 | 316.8 / 177.0 | 4103 |



### 4.2 Outer wind speeds

The QSCAT-R database was used to validate the computed (outer) azimuthal wind speeds while using the H10 wind profile in combination with several sources to constrain the decay of wind speeds. QuikSCAT includes 690 unique tropical cyclones and is known to provide reliable results for outer wind speeds of lower intensity. Figure 7 displays the error

profile, representing the difference between modeled wind speed and measured data based on QuickSCAT, as a function of the normalized radius. This means that for all validated TCs the radius on the x-axis is divided by the RMW. A horizontal line equal to zero indicates no difference between modeled and measured wind speed data, while the solid colored lines represent the median difference. The filled area indicates the interquartile range (IQR).

The figure shows that in combination with the H10 wind profile the proposed relationships results in the smallest

difference with respect to the measured wind speeds (green line). However, applying H10 wind profile with observed values for the wind radii (i.e. based on BTD values) results in an underestimation of the modeled outer winds (blue line). On the other hand, applying the H10 wind profile, without additional information on the gale force winds (H80), results in a strong overestimation of the outer winds (red line). Similarly, a combination of other existing relationships for RMW (VW08) and ΔAR35 (CLIPER) results in an overestimation of the outer winds but to a lesser degree (orange line).

The same information is also shown in Table 5, where the root-mean-square differences and bias between modeled wind speeds and measurements are summarized. Using the proposed relationship with the H10 wind profile results in the lowest RMSD and smallest bias.

**Table 5** **Root-mean-square difference (RMSD) and bias (in m/s) between modeled and measured azimuthally averaged**

**wind speeds based on QSCAT-R data. The data analysed in the table refer all TCs with wind speeds between 40 m/s and 5 m/s and a normalised radius between 3 and 16. Statistics are shown for median values (50%) and the IQR range (25%-75%). With 'H10: observed' the authors refer to the Holland et al. (2010) wind profile in combination with the RMW and AR35 from the BTD.**

| Wind profiles | RMSD median (50%) | RMSD: low (25%) | RMSD: high (75%) | bias: median (50%) | bias: low (25%) | bias: high (75%) |
|---|---|---|---|---|---|---|
| H80 | 11.24 | 8.32 | 14.57 | 10.98 | 7.89 | 14.34 |
| H10: observed (BTD) | 5.46 | 3.85 | 7.04 | -4.67 | -6.32 | -2.6 |
| H10: VW08 + CLIPER | 3.60 | 2.06 | 5.76 | 1.64 | -1.27 | 4.16 |
| H10: proposed | 2.86 | 1.71 | 4.51 | -1.04 | -3.3 | 1.39 |


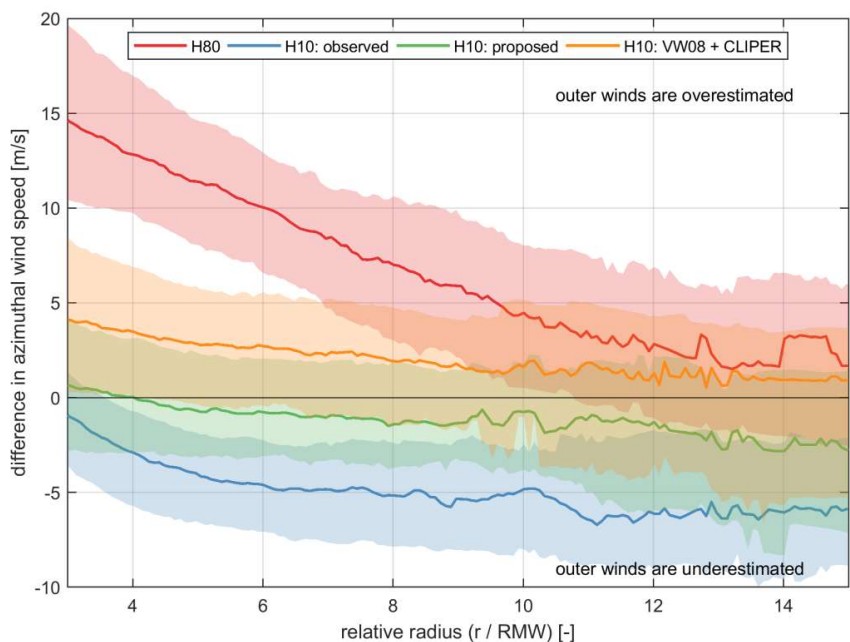

**Figure 7** **Wind speed error profiles from different models versus observations from QuikSCAT as function of r / RMW.**
**A value equal to zero indicate a perfect match between model and observations. Interquartile ranges are shown with filled colors.**
**Note: for the proposed relationships the most probable value for RMW and AR35 was used (i.e. mode).**



## 5. Discussion

For clarity, discussion points have been grouped under three main topics:

### 5.1 Data

In this study, all available BTD from NHC and JTWC were used and combined into one dataset. This approach was
followed to create the largest sample size possible, in order to derive empirical (stochastic) relationships valid for each ocean basin, various latitudes, different TC geometries and strengths. This approach is limited by the debatable assumption that each six-hourly data point is statistical independent. Moreover, errors in the BTD can be quite significant so previous studies (e.g. Holland, 2008) selected a specific subset of the BTD in order to ensure the quality of the data and remove potential inconsistencies. However, the advantage of including all data entries is that the derived relationships are more widely
applicable (i.e. larger parameter space). Moreover, as they are based on larger datasets, they treating TC geometry variables to be treated in a stochastic rather than a deterministic approach.

### 5.2 Methodology

In order to derive the new empirical relationship for RMW and $\Delta AR35$, the maximum sustained wind speed and latitude were used. Although other authors used additional parameters to describe the TC geometry (e.g. pressure drop, storm
duration, rapid intensification, etc.), limited predictive skill improvement was found by incorporating those additional parameters. This makes the derived relationships relatively simple for practical applications. Moreover, lognormal statistical distributions in combination with exponential functions were used to fit all available data and derive those relationships. For our application, exponential-shaped functions resulted in the best fit compared to the available data. The choice of lognormal statistical distributions was based on the comparison of the different CDFs derived using different distributions, the
Kolmogorov-Smirnov test and supported by findings from literature (e.g. Dean et al., 2009; Chavas et al., 2016). However, different statistical distributions and functions are available in literature to fit and describe TC geometry data. The strength of using statistical distributions to derive these relationships is that TC geometry is treated stochastically, therefore providing not only mean/median values but also prediction intervals. This is especially of importance when the TC geometry is not known (e.g. for older BTD and Monte Carlo analysis with synthetic tracks) with numerical models. Another possibility
would be the derivation of wind speed probability estimates.

### 5.3 Differences in measured and modeled outer wind profiles

QuikSCAT data were used to validate the (outer) azimuthal wind speeds derived using the new empirical relationships in combination with the H10 wind profile. The analysis has shown how the proposed relationships in combination with the H10 wind profile result in the lowest RMSD and smallest bias for the outer winds, compared to other



existing relationships (see Figure 7). This gives confidence that parametric wind models can be used to compute the outer wind speeds. This is of particular importance for the estimation of coastal hazards (i.e. storm surge and wave heights).

However, differences were also found for individual TCs, where the Holland wind profile in combination with the empirical relationships derived in this paper did not result in a good reproduction of the outer wind speeds. As an example,

Figure 8A shows computed and measured wind speeds for TC Vaianu (2006), which was characterized by an extremely large radius of gale force wind (R35 equal to 292 km ≈ 10% probability of exceedance). Measured values are shown in the figure by the black circles. When applying the proposed relationships to compute the most probable values of the wind radii (red line), a R35 value equal to 162 km is obtained, resulting in an overall underestimation of the measured outer wind speeds. Also, when using the observed wind radii information (blue line), TC outer winds are not well reproduced, which

shows that even with the correct wind radii value, parametric wind models can have the wrong shape. This approach is also limited when measured wind speeds cannot be represented by an exponential decay, as it is assumed by the Holland wind profile. For example, TCs characterized by two wind maxima cannot be reproduced by an exponential decay of wind speed (Figure 8B. However, the Holland wind profile is widely used due to its relative simplicity and does, most of the times (80% of the TCs are reproduced with a RMSE of less than 5 m/s), reproduce the decay of wind speed fairly well as shown in the

evaluation of 690 unique TCs in Figure 7.

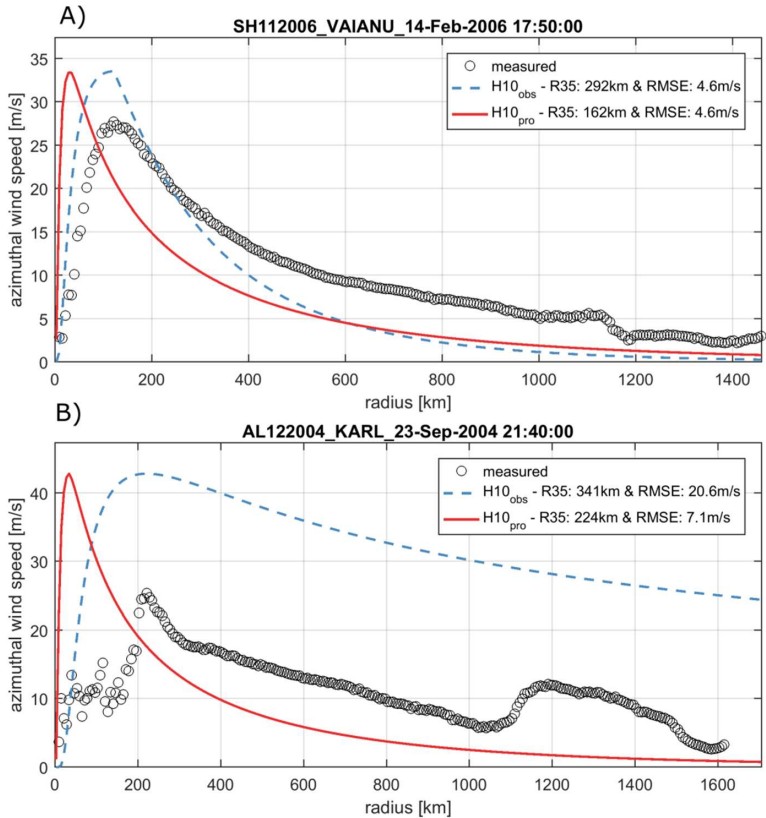

**Figure 8** Radial wind profiles for measured (black circles), computed based on relationships for wind radii (red lines) and computed based on observed wind radii for tropical cyclone Vaianu (14 February 2006) (panel A) and Karl (23 September 2004) (panel B). Measured data are based on QSCAT-R data, while computed values are based on H10 calibrated with the relationships proposed in this paper or observed data. Panel A and B are specific examples indicating when difference between measured wind speeds and TC size can be encountered.


## 6. Conclusions

In this paper, new empirical relationships are derived which estimate tropical cyclone (TC) geometry with simple and generic equations and with higher accuracy with respect to other well-known empirical relationships available from literature. Those new relationships are valid for any ocean basin (Atlantic, S/NW/NE Pacific, N/SE/SW Indian Ocean).

Moreover, the new relationships include a stochastic description for both the radius of maximum winds (RMW) and the radius of gale force winds (ΔAR35). This allow the quantification of the prediction interval around the median estimates, making the estimation more useful.

According to the derived relationships, the RMW is described as a function of the maximum sustained wind speeds and latitude. The radius of gale force winds is estimated using a newly introduced ΔAR35 parameter (average difference

between radius of 35 knots and radius of maximum wind), and is also dependent on the maximum sustained winds and latitude. Both parameters are fit through simple exponential functions. Compared to best track data, the proposed relationships improve the estimation of RMW and ΔAR35 by reducing the root mean square difference (RMSD) up to 25%. Larger improvements were found in particular for non-US TCs, since most of the existing relationships are based on data from the Atlantic Ocean, Northeastern Pacific Ocean and/or Northwestern Pacific Ocean.

The new relationships, in combination with the Holland wind profile, were validated using a subset of the BTD and (outer) azimuthal wind speeds from the QSCAT-R database. The results showed that (outer) azimuthal wind speeds of the TC can be reproduced with the H10 wind profile when using either the BTD ('observed') for RMW and ΔAR35 or the relationships derived in this paper. When no additional information on wind radii was used to calibrate the H10 wind profile, which is generally done when the radius of gale force wind is not known, surface wind speeds were overestimated.

The derived empirical relationships can be used in a variety of applications. For example, a better estimate of TC pressure and surface wind speeds for Monte Carlo analysis with synthetic tracks for risk assessments with numerical models can result in a more accurate description of wave and surge conditions resulting from the TC. As a result, this can lead to a better quantification of coastal hazards, and consequent risks and damages. Similarly, an improved assessment of those hazards can help the design of appropriate adaptation measures. Other fields of applications may vary from improved design

parameters for offshore structures to navigation. The application of the new empirical relationships will be presented as part of a separate paper currently under preparation.

*Acknowledgments.*

We are thankful to the Deltares research programs "Extreme hydro and morphodynamic events" and 'Enabling

Technologies' which has provided financings to write this paper.



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
