# Peer review of "Estimates of tropical cyclone geometry parameters based on best track data"

_Natural Hazards and Earth System Sciences, 2019_

## Referee Comment (RC1) · Anonymous Referee #1 · 7 Jul 2019

The authors use Best Track data to improve existing formulas for tropical cyclone characteristics, treating the variables as stochastic ones. The paper is well written and without doubt presents a useful step forward in the field. The approach followed is clear and the authors also share with the community the new parameterizations they produced. Overall I have no objections on the publication of the paper, just few comments/suggestions mostly for consideration.

- Given the background of the authors, the paper is currently oriented to ocean modelers, however the work could be useful also to anyone dealing with TC hazards. To that direction I would recommend expanding a bit the introduction also to wind hazards, adding more references (e.g. Peduzzi 2012 Nature Climate Change). On the ocean modeling side I would recommend citing some recent papers simulating tropical cyclones (e.g. Bloemendaal 2019, Climate Dynamics; Vousdoukas 2018, Nature Communications)

- One major weakness of the study is that BTD are not accurate and as also shown in the validation. I think this is made clear in the discussion by the authors, but I was wondering whether the performance would be further improved if one could consider the BTD error in the fitting of the empirical equations (thinking of the possibility of introducing the error maybe by doing the least-square fitting in a Monte Carlo framework). I leave it to the authors if they would like to discuss about it in the paper or in the present open discussion. In any case, I would like to see the authors thoughts on how the parameterizations could be further improved; as it is clear that despite the obvious progress the data to provide a satisfactory solution to the problem are still not there.

- The correlation implied by figs 3 and 5 is ...daunting! I would suggest the authors to provide some additional information: for example RMSE and bias expressed as %, r2 coefficients, but also some q-q plots (or scatter plots with colorscale expressing point density) which could show that despite the scatter the two variables are somehow related. For the time being especially fig 5 seems like noise.

- Some figure captions could benefit from better explanations of the contents of the figure (especially 7-8)

---

## Referee Comment (RC2) · Anonymous Referee #2 · 15 Aug 2019

General comments: This study proposed empirical relationships to estimate RMW and R35 based on BTD data. The results are promising since they compare better with QSCAT-R data than other approaches in the literature. TC geometry or radial wind fields are critical for TC damage estimate. In this sense, the study is useful and will be a good add to the TC hazard community. The manuscript is overall well-written and organized. However, some of the presentation can be improved and clarified, e.g., some figures and their captions.

Specific comments: 1. What is A1 in Table 1? It has not been defined and referred to. 2. How do the authors come up with the exponential function forms (Eq. 2 and 4)? 3. P8L1, the sentence starting with "a shape parameter..." is rather confusing to me. 4. P4L9, why delta_AR35 contains less scatter? It is just the difference of R35

from RMW, right? If this is correct, it might imply a close connection between R35 and RMW. This can be mentioned and discussed. 5. References might need to be checked for the format requirement of the journal.
* * *

---

## Author Comment (AC1) · 21 Sep 2019

Dear editor, dear reviewers,

On the 10 of April 2019, we have submitted the following manuscript to the Journal of Natural Hazards and Earth System Sciences titled:" Estimates of tropical cyclone geometry parameters based on best track data " (MS No.: nhess-2019-119). On the 26st of August 2019, we were informed that the open discussion was completed. In total, we received comments by two reviewers which provided a very positive feedback on the work done and valid suggestions. With the message we would like to acknowledge their time and efforts which we believe have improved the quality of our manuscript. Below you will find a reply to all the specific questions and suggestions which have

also been implemented in the original manuscript.

Kind regards, Kees Nederhoff

— Reviewer #1:

General Comments: 1. The authors use Best Track data to improve existing formulas for tropical cyclone characteristics, treating the variables as stochastic ones. The paper is well written and without doubt presents a useful step forward in the field. The approach followed is clear and the authors also share with the community the new parameterizations they produced. Overall I have no objections on the publication of the paper, just few comments/suggestions mostly for consideration

Thank you very much for these kind words, we also believe that this work is an important addition to our field!

2. Given the background of the authors, the paper is currently oriented to ocean modelers, however the work could be useful also to anyone dealing with TC hazards. To that direction I would recommend expanding a bit the introduction also to wind hazards, adding more references (e.g. Peduzzi 2012 Nature Climate Change). On the ocean modeling side I would recommend citing some recent papers simulating tropical cyclones (e.g. Bloemendaal 2019, Climate Dynamics; Vousdoukas 2018, Nature Communications)

We agree with the reviewer. We have extended the literature review including the abovementioned references (page 1 lines 26-28 and page 2; lines 3-4).

3. One major weakness of the study is that BTD are not accurate and as also shown in the validation. I think this is made clear in the discussion by the authors, but I was wondering whether the performance would be further improved if one could consider the BTD error in the fitting of the empirical equations (thinking of the possibility of introducing the error maybe by doing the least-square fitting in a Monte Carlo framework). I leave it to the authors if they would like to discuss about it in the paper or in the

present open discussion. In any case, I would like to see the authors thoughts on how the parameterizations could be further improved; as it is clear that despite the obvious progress the data to provide a satisfactory solution to the problem are still not there.

Reviewer #1 is making a very good point, and, in fact, we did explore this path before settling on the parent distributions idea. BTD error fitting results in limited improvement. Moreover, there is limited literature supporting this idea. This in contrast to testing several parent distributions. We have extended the discussion section on this and included our thoughts on how to further improving the relationships (page 20; lines 25-27).

4. The correlation implied by figs 3 and 5 is ...daunting! I would suggest the authors to provide some additional information: for example RMSE and bias expressed as %, r2 coefficients, but also some q-q plots (or scatter plots with colorscale expressing point density) which could show that despite the scatter the two variables are somehow related. For the time being especially fig 5 seems like noise

The correlation in Figure 3 and 5 shows indeed the need for new TC geometry relationships and the stochastic nature. We have extended the discussion on the error statistics and added the RMSE and bias as percentage as part of the discussion (page 7; lines 19-21 and page 12; lines 18-19). We feel that adding q-q plots for the relationships in literature is outside the scope of this paper since we want to focus on the improved relationships based on BTD and the stochastic nature of TC geometry.

5. Some figure captions could benefit from better explanations of the contents of the figure (especially 7-8)

In the current version of the manuscript, we have improved the explanation for all the figures and specially for Figure 7 (page 19) and Figure 8 (page 22).

Reviewer #2

General comments

1. General comments: This study proposed empirical relationships to estimate RMW and R35 based on BTD data. The results are promising since they compare better with QSCAT-R data than other approaches in the literature. TC geometry or radial wind fields are critical for TC damage estimate. In this sense, the study is useful and will be a good add to the TC hazard community. The manuscript is overall well-written and organized. However, some of the presentation can be improved and clarified, e.g., some figures and their captions

Thank you very much for these kind words, we also believe that TC geometry is an important aspect to get more reliable TC damage estimates. We have made improvements to the captions across the manuscript (also in agreement with comment by Reviewer #1 comment 5)

Specific comments

1. What is A1 in Table 1? It has not been defined and referred to.

The reviewer is correct: A1 is simply the shape parameter used in the fitted lognormal coefficient. We have removed A1 from Table 1 and changed the sentence explaining how the fit was done (page 8; lines 4-5)

How do the authors come up with the exponential function forms (Eq. 2 and 4)?

As explained in the discussion section (page 20; lines 13-27) the use of exponential functions, was essentially based on insight obtained during the pre-analysis of the data.

2. P8L1, the sentence starting with "a shape parameter: : :" is rather confusing to Me

In the current version of the manuscript, we have improved this sentence (page 8; lines 4-5)

3. P4L9, why delta_AR35 contains less scatter? It is just the difference of R35 from RMW, right? If this is correct, it might imply a close connection between R35 and RMW. This can be mentioned and discussed.

That is correct, AR35 is computed by subtracting RMW from R35. The point of the reviewer has now been mentioned and briefly discussed on page 4 lines 14-15.

4. References might need to be checked for the format requirement of the journal.

To the authors understanding, the NHESS requires this type of references since the Bibtex Bibliographic Style File has been used to generate the reference list. However, we are happy to review this, if necessary.
* * *

---

## Author Comment (AC3) · 21 Sep 2019

See the attached supplement

Please also note the supplement to this comment:
https://www.nat-hazards-earth-syst-sci-discuss.net/nhess-2019-119/nhess-2019-119-AC3-supplement.zip
* * *

---

## Author Comment (AC4) · 1 Oct 2019

Dear editor, dear reviewers,

On the 21 of September 2019, we have submitted the rebuttal letter regarding the manuscript 'Estimates of tropical cyclone geometry parameters based on best track data'. However, it has come to our attention that in that manuscript, two typos were present. - In Equation 4, we flipped the sigma and mu symbol. - Figure 6 should be AR35 instead of delta AR35. Also, we have changed a few reference to this figure.

In the attached manuscript we have corrected these typos.

Kind regards,

[Figure]

Kees Nederhoff

Please also note the supplement to this comment:
https://www.nat-hazards-earth-syst-sci-discuss.net/nhess-2019-119/nhess-2019-119-AC4-supplement.zip